# Copaiba Oil-Based Emulsion as a Natural Chemotherapeutic Agent for the Treatment of Bovine Mastitis: In Vivo Studies

**DOI:** 10.3390/pharmaceutics15020346

**Published:** 2023-01-20

**Authors:** Katieli da Silva Souza Campanholi, Ranulfo Combuca da Silva Junior, Flávia Amanda Pedroso de Morais, Renato Sonchini Gonçalves, Bruna Moura Rodrigues, Magali Soares dos Santos Pozza, Lidiane Vizioli de Castro-Hoshino, Silvio Mayke Leite, Otávio Augusto Capeloto, Mauro Luciano Baesso, Paulo Cesar Pozza, Wilker Caetano

**Affiliations:** 1Chemistry Department, State University of Maringá, Maringá 87020-900, Brazil; 2Centre de Recherches sur les Macromolecules Végétales (CERMAV)-CNRS, Université Grenoble Alpes, 601, rue de la Chimie, BP 53, CEDEX 9, 38041 Grenoble, France; 3Laboratory of Chemistry of Natural Products, Department of Chemistry, Center for Exact Sciences and Technology, Federal University of Maranhão, São Luís 65080-805, Brazil; 4Animal Science Department, State University of Maringá, Maringá 87020-900, Brazil; 5Physics Department, State University of Maringá, Maringá 87020-900, Brazil; 6Institute of Health and Biotechnology of Coari, Federal University of Amazonas, Coari 69460-000, Brazil

**Keywords:** bovine mastitis, copaiba oil-resin, emulgel, natural anti-inflammatory, photoacoustic, permeation study

## Abstract

Copaiba oil-resin (COR) extracted from *Copaifera reticulata* Ducke has been used as a natural chemotherapeutic agent for a wide range of therapeutic applications. This study presents an emulgel design with a high concentration of COR, designed to prevent and treat mastitis. The COR was stabilized in a gel matrix constituted by carbopol C934P and Pluronic^®^ F127 (ECO formulation) ratios. The permeation study of ECO was accessed by Fourier transform infrared photoacoustic spectroscopy (FTIR-PAS). The results reveal a high capacity of ECO to permeate deep skin layers. Dairy cows with a history of mastitis were used as in vivo models and exposed to ECO treatment. Monitoring of the teat’s inflammatory response showed that ECO effectively prevents mastitis. Furthermore, the ECO formulation was able to form a thin film gel on the application side, preventing fly proliferation and significantly reducing the pathogen load. This study reveals a drug that can used as an alternative application for mastitis in human or veterinary clinics.

## 1. Introduction

Mastitis is a potentially severe illness that leads to inflammation of the breast tissue, affecting 33% of breastfeeding women. One consequence of mastitis is reduced breastfeeding time, besides significant macronutrient content changes that reduce milk quality. The increasing incidence of etiological agents’ resistance to current antibiotics has caused concern in clinical practice. Thus, the scientific community are forced to search for new therapies as alternative treatments [1].

In the dairy industry, mastitis also represents an economically costly disease since it reduces the productive life of the affected cows and the quality of the milk. Among the issues that lead to bovine mastitis is poor hygiene in dairy cows. This contributes to the proliferation of pathogens [2]. Although many synthetic drugs have been developed to treat mastitis, the impact of mastitis on milk production remains economically significant. Products that act as safe microbicides and healing agents, without leaving residues or impairing the milk’s physical and chemical quality, are needed to treat this pathology [2,3,4]. 

Copaiba oil-resin (COR) has been highlighted as a natural anti-inflammatory for treating skin diseases [5,6]. Studies show benefits in the use of COR for mastitis treatment due to its broad-spectrum antibiotic effect [4,6,7,8,9,10]. Furthermore, COR has a healing [11,12] and tissue repair effect [13], which encourages the development of COR formulations for topical therapies. 

In this work, we present the use of a copaiba oil-based phytotherapeutic emulgel as a proposal for human and veterinary clinics. The formulation offers the differential of supporting 20% COR for prolonged periods without phase separation. Other articles have reported formulations with low COR contents due to the phase separation process. The administration of ECO allowed the formation of a protective film that was capable of barring the entry of pathogenic microorganisms into topical lesions or teats. This behavior is promising, because the teat sphincter remains open for hours after milking, and a physical and chemical barrier is fundamental for the animal’s protection. Furthermore, the formulation uses low polymer concentrations, which makes the drug inexpensive and accessible to farmers or low-income populations. These results given in this paper are based on in vivo studies conducted on lactating cows with a history of mastitis. The results show the potential of ECO in mastitis prevention and the availability of chemotherapeutic agents in deep layers of the skin.

## 2. Materials and Methods

### 2.1. Materials

Carbopol C934P^®^ was purchased from Lubrizol Advanced Materials (Rio de Janeiro, Brazil), and Pluronic^®^ F127 (MW = 12,600 g·mol^−1^), and triethanolamine (TEA) were obtained from Sigma Aldrich (São Paulo, SP, Brazil). The lactic acid was of the Ekomilk pos film gel brand, bought in the local market (Paraná, Maringá, Brazil), and applied as purchased. The sterile swab for sample collection (3M™ Quick Swabs) was obtained from Biomed micro biotechnology (Minas Gerais, Belo Horizonte, Brazil), and the plate count agar, PCA DIFCO^®^, was acquired from Induslab (Paraná, Londrina, Brazil). All experiments were conducted using purified water obtained from the Milli-Q system (Millipore, Merck, Darmstadt, Germany).

### 2.2. Extraction Process

The COR was obtained from the Copaiba da Amazonia company. The oil was obtained from trees from the rainforest of the agro-extractive association reserve (Apuí, Amazonas, Brazil). The COR was collected by employing a traditional manual method, using a 1-inch drill to collect approximately 1 L of oil from each tree. After collection, the holes were closed, respecting the cyclic periods used for extraction (3 years). The collection of bioactive material complied with the environmental laws in vigor (National System for the Management of Genetic Heritage—SISGEN nº AE28797; and the System of Authorization and Information of Biodiversity—SISBIO nº 72922-1).

Geolocation information: The coordinates of the study area, Amazonian (Apuí), where the oil-resin sample was collected are 7°55′28.5″ S latitude, 60°15′10.7″ W longitude. The in vivo studies were conducted in Parana (at the Experimental Farm of Iguatemí belonging to the State University of Maringá, Brazil), located at the following coordinates: 23°21′36.83″ S latitude and 52°04′27.63″ W longitude.

### 2.3. ECO Preparation

The ECO formulation was prepared by dispersing 1.2% *w*/*w* Carbopol C934P and 2.4% *w*/*w* Pluronic^®^ F127 in water under vigorous stirring until complete homogenization occurred. Then, the pH was adjusted to 7 using TEA. Afterward, COR was slowly added until it reached 20% *w*/*w*, and the system was continuously stirred for 30 min (Figure 1). After manufacturing, the emulgel (called ECO, a gel containing COR) was stored at room temperature. The details of the optimization and preparation have recently been published [12].

### 2.4. Apparent Viscosity Study

The ECO, COR, and Ekomilk pos film gel (commercial gel) viscosity were measured via a B-one plus viscometer (Lamy Rheology instrument, Champagne-au-Mont-d’Or, France), with 1 min of analysis time under 250 rpm/min rotation, at 25 °C. All measurements were carried out using four replicates.

### 2.5. Skin Permeation Studies of COR and ECO

The porcine ear segments used in this study were provided by the slaughterhouse of the Experimental Farm of the State University of Maringá (with authorization from the Municipal inspection service). Skin segments with white coloration and without hematomas from young and recently slaughtered animals were selected. The samples were cleaned with ultrapure water and dissected to remove the cartilage and subcutaneous fat [14]. Approximately 30 mg of ECO and COR was administered on the stratum corneum side of the skin in an area of approximately 1 cm^2^. After 30 min, the spectra were obtained by Fourier transform infrared photoacoustic spectroscopy (Vertex 70v, Bruker Optik GmbH, Ettlingen, Germany), a PAS (PA 301, Gasera Ltd., Turku, Finland) between 3800 and 1000 cm^−1^, a resolution of 8 cm^−1^, and 16 scans [15]. The photoacoustic cell was purged with helium gas before starting each measurement. The determination of the depth of the skin sample that contributed to the photoacoustic signal was performed by a previously documented methodology [14]. The frequency used in the experiments was 500 Hz. The spectra obtained have been subjected to baseline correction and normalization.

### 2.6. Case Study: ECO Potential in Mastitis Prevention

The experiments were approved by the Animal Ethics Committee of the State University of Maringá under number—CEUA n° 1234080221. Six Dutch breed dairy cows from the herd at the Experimental Farm of Iguatemí (in the district of Maringá, Brazil), belonging to the State University of Maringá, were used. At the end of lactation, the animals presented a history of subclinical or clinical mastitis (evidenced by the CMT—California mastitis test, and the black mug test). The experimental procedure was performed for 14 days, with treatment after milking. Three animals received post-dipping applications with the positive control (lactic acid, 10 mL in each teat using non-return plastic liners), and three animals were treated with the ECO product (approximately 2 g per teat, manually administered) [2]. The small number of animals was justified by the small dairy cattle production scale at the university’s experimental farm.

Microbiological evaluations and milk analysis were performed on the 1st, 7th, and 14th days. Analysis of mesophilic aerobic bacteria was performed by collecting swabs from the teats and seeding on PCA (plate count agar) (4 teats of the animals constituted one sample). The milk from each teat of each animal was collected individually in the same sterilized glass container. The first three jets were discarded, and 100 mL of the subsequent ones were stored. The milk somatic cell count (SCC) was collected using aliquots from the milking gallon after homogenization. The collected milk was kept at 4 °C and immediately analyzed. The total mesophilic aerobic bacteria count was determined using PCA agar for the milk, keeping the incubation at 35 °C for 48 h. The analyses of subclinical or clinical mastitis incidence considered the SCC (EKO Milk Scan, CapLab, São Paulo, Brazil). The microbiological results were submitted to the *t*-test using Prism 5.0 software (Graph, San Diego, CA, USA), considering a 5% significance level [2].

## 3. Results and Discussion

ECO presented a homogeneous and white aspect. Besides, ECO did not show phase separation principles for a period of more than nine months. At pH 7, the carboxylic acid groups of the Carbopol chain are ionized, and the consistency of the ECO is considerably increased [16]. This behavior allowed a stabilization effect, arising from the absence of oil droplet mobility (emulsion-filled gel formation, Figure 1), increasing the temporal stability of the system. During the preparation stage, the high shear rate reduced the size of the emulsion droplets, which slowed down the coalescence processes [17]. 

We have recently shown all of the rheological and mechanical characterizations for the ECO formulation, in addition to the stability studies required by the national guidelines for product registration. The ECO formulation exhibits behavior typical of semi-solid systems, with adhesive, pseudoplastic, and viscoelastic properties [12]. We also show in vivo studies that demonstrate the healing, larvicidal and antibiotic potential for this formulation [12].

The absolute viscosity (Table 1) of ECO has increased 45-fold more than COR and 13-fold more than lactic acid (commercial gel). This behavior is an essential property, as the increased consistency favors the longer duration of the ECO formulation at the treatment site. 

The FTIR-PAS spectral comparison between the COR and ECO allows for the identification of the characteristic oil-resin peaks after ECO obtention (Figure 1). This information indicates the stabilization of the fixed and volatile components in the polymer matrix [18].

The infrared spectra of COR showed the typical natural frequencies of vibration, being in the region between 2870 and 2931 cm^−1^, which is appropriate for terpenes and other organic compounds. The additional vibrations (Table 2) agree with the spectra shown in the literature [19,20]. ECO spectra showed COR characteristic signals (Figure 1) without considerable shifts. This result indicates that the main components of the oil-resin were preserved after ECO obtention.

The FTIR-PAS analysis of the ECO constituents served as a tool to evaluate their skin permeation potential (Figure 2). Porcine skin was selected as a model system for skin permeation due to its morphological similarity to human skin [14,24].

The epidermis’ contact with COR allowed the cutaneous partitioning of the natural chemotherapeutic agents, as verified by the oil-resin characteristic signals (from 1500 to 1100 cm^−1^) on the dermis side (Figure 2A). ECO administration also revealed a cutaneous partitioning of the oil-resin therapeutic components. Considering that the ECO formulation contains 20% of COR mass, the similarity between the COR and ECO spectra (Figure 2A) demonstrates the efficiency of ECO in releasing the oil-resin constituents at higher concentrations than COR, as shown by the insert in Figure 2A. These behaviors, and the C934P/F127 mixture’s signals on the dermis, indicate that the polymer blend acted as a skin permeation promoter [10,25,26]. The emollient properties of the oil-resin also favor cutaneous permeation by intrinsic skin mechanisms. The measured average thickness of the skin was 809 µm. The calculated thermal diffusion length showed values between 7 and 14 µm of thickness, corroborating the FTIR-PAS analysis and related literature [27]. 

Additionally, ECO demonstrated anti-inflammatory potential through in vivo studies. ECO administration on the ceilings was easy. Furthermore, ECO spreading leads to a protective thin film over the teats (Figure 3), which acts as a natural repellent against fly proliferation (myiasis) [28]. 

The total count of aerobic mesophiles and somatic cells in raw milk (Table 3) prove the therapeutic potential of ECO for mastitis.

A statistical comparison between ECO and the control gel treatment (lactic acid) showed an equivalence in the results (*p* > 0.05). The high milk quality during ECO treatment was confirmed by the low mean of aerobic mesophilic/milk (AMM) bacterial counts (3.46 log10/mL), which remained below the value recommended by the normative instruction (NI) N°76 (5.47 log10; 300.000 CFU/mL) [29].

The aerobic mesophilic/swab (AMS) bacteria count collected before milking remained below for both products. Similarly, there was no significant difference (*p* > 0.05) between the treatments during the application period for this count. This result shows that the udders of the dairy cattle did not act as reservoirs of pathogens. Under these conditions, the transmission effects during milking would not be significant and would not result in chronic subclinical infections [2,3].

No significant difference (*p* > 0.05) was observed to the somatic cell count (SCC) after the ECO treatment in comparison to the control (5.68 log10; 500.000 SC/mL). Moreover, the values obtained were lower than the current regulations, which establish a maximum count of 5.70 log10 (500.000 SC/mL) by NI 76 [29]. The SCC high values obtained by ECO treatment were influenced by the age and history of the animal, which shows an average of SCC values in log10/mL of 5.77 ± 0.15 (monthly milk control history data, with a coefficient of variation of 3% for the last eight months before treatment). The age, duration of the infectious condition, and a high history of SCC of the animal, reduce the probability of cure in cases of diagnosis. Therefore, preventive treatments and the use of drugs that act as a control are fundamental for an animal’s life quality as well as for the milk produced. These results demonstrate the potential for formulations containing copaiba oil-resin in mastitis treatment.

## 4. Conclusions

The ECO formulation allowed for the formation of a protective film on the teats, and released the chemotherapeutic agents into deep layers of the skin. Furthermore, the ECO treatment maintained a low mesophilic aerobic bacteria count in both the milk and teats, preventing mastitis. Likewise, the somatic cell count showed values within acceptable standards, considering the history of the animal. Furthermore, the ECO formulation had the same efficiency as the commercial product (lactic acid gel). Thus, ECO is a natural anti-inflammatory agent for the preventive treatment of dermatological diseases. It offers an alternative for treating or preventing mastitis, either in human or veterinary clinics.

## Data Availability

Not applicable.

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
