# Peer review of "Copaiba Oil-Based Emulsion as a Natural Chemotherapeutic Agent for the Treatment of Bovine Mastitis: In Vivo Studies"

_pharmaceutics, 2023, doi:10.3390/pharmaceutics15020346_

Round 1
Reviewer 1 Report
In the paper "Design of an emulgel based on copaiba oil-resin as a natural chemotherapeutic agente for the treatment of bovine mastitis" an application of COR for mastitis treatment. The manuscript is short, however this is normally for communications type. Below I have several comments:
1. Why the spectra resolution and number of scan of FTIR measurement was so low?
2. What type of normalization and baseline correction was used in analyze of FTIR spectra?
3. The discussion section was poor.
Author Response
We greatly appreciate your correction. All your requests have been complied with and your questions answered.
Thank you and have a blessed year 2023.
My best Regards
Katieli

Reviewer 2 Report
This manuscript describes an emulgel based on copaiba oil resin. It is interesting. However, reviewer thinks further description and discussion are required in some points.
1. The title of the communication mentions "Design of an emulgel," but formulation study of emulgel has not been conducted, and information on the properties of emulgel is a short description of the stability of phase separation and viscosity on just one emulgel formulation. The size of droplet and viscosity of formulations are factors that affect the release and permeability of oil resin from formulations, and they vary depending on the formulation. Reviewer would like to see a description of the formulation design.
2. In the introduction, please mention the scientific novelty. Please describe the problem to be solved in this research and why emulgel has been selected to be discussed in this communication.
3. Although the permeability of oil resin from emulgel is suggested in Fig. 2 and Table 3, there is no quantitative information on the permeability. Therefore, reviewer cannot understand whether the formulation of emulgel under consideration in this study works as intended. Reviewer requests to describe whether the permeability changes if the formulation of the emulgel is changed.
Author Response
We greatly appreciate your correction. All your requests have been complied with and your questions answered.
Thank you and have a blessed year 2023.
My best regards
Katieli

Round 2
Reviewer 2 Report
The authors appropriately respond to reviewers’ comments and suggestions.
The points authors want to make in this communication are now clear.
I think this manuscript is now acceptable for publication.